# Efficient protein production by yeast requires global tuning of metabolism

Mingtao Huang[1,2], Jichen Bao [1,2], Björn M. Hallström[3], Dina Petranovic[1,2] & Jens Nielsen [1,2,3,4]

The biotech industry relies on cell factories for production of pharmaceutical proteins, of which several are among the top-selling medicines. There is, therefore, considerable interest in improving the efficiency of protein production by cell factories. Protein secretion involves numerous intracellular processes with many underlying mechanisms still remaining unclear. Here, we use RNA-seq to study the genome-wide transcriptional response to protein secretion in mutant yeast strains. We find that many cellular processes have to be attuned to support efficient protein secretion. In particular, altered energy metabolism resulting in reduced respiration and increased fermentation, as well as balancing of amino-acid biosynthesis and reduced thiamine biosynthesis seem to be particularly important. We confirm our findings by inverse engineering and physiological characterization and show that by tuning metabolism cells are able to efficiently secrete recombinant proteins. Our findings provide increased understanding of which cellular regulations and pathways are associated with efficient protein secretion.

[1] Department of Biology and Biological Engineering, Chalmers University of Technology, SE41296 Gothenburg, Sweden. [2] Novo Nordisk Foundation Center for Biosustainability, Chalmers University of Technology, SE41296 Gothenburg, Sweden. [3] Science for Life Laboratory, KTH Royal Institute of Technology, SE17165 Solna, Sweden. [4] Novo Nordisk Foundation Center for Biosustainability, Technical University of Denmark, DK2800 Kongens Lyngby, Denmark. Correspondence and requests for materials should be addressed to J.N. (email: nielsenj@chalmers.se)

Eukaryal cells have a sophisticated protein secretory system, which ensures proper protein folding, post-translational modification, sorting, trafficking, etc[1–3]. Many other cellular processes interact closely with the protein secretory pathway to ensure supply of building blocks and energy[4]. For this reason, dysfunction of the protein secretory pathway can be lethal to the cell, and indeed many human diseases result from disorders in this pathway[5, 6]. Yeast *Saccharomyces cerevisiae* is a single-cell organism that is widely used as a model to study eukaryal cell biology, including the protein secretory pathway. Indeed, large knowledge about protein secretion in eukarya has been obtained from studies of this yeast[7–9]. Yet, full understanding of the architecture of this pathway and in particular its interaction with other cellular processes is still lacking.

Eukaryal cells are often preferred cell factories for production of many pharmaceutical proteins as they can be engineered to secrete functional proteins with correct fold and modifications into the extracellular medium, which results in reduced costs for downstream purification[10]. Mammalian cells, insect cells, filamentous fungi, and yeasts are, therefore, widely used cell factories for production of recombinant proteins[11]. Many studies have focused on improving protein secretion of these cell factories through metabolic engineering by elimination of bottlenecks at different steps, especially in the secretory processes[12–15]. However, limited understanding of the protein secretory pathway prevents rational engineering of many of these cell factories. There is, therefore, a need for unravelling the underlying mechanisms, and in particular how the secretory pathway and its regulation interact with other cellular processes. We use RNA-seq to perform a transcriptional genome-scale analysis of seven mutant strains of the yeast *S. cerevisiae* having a fivefold varying protein secretion capacity for a recombinant protein. Higher protein secretion may be affected not only by the direct process that a mutant gene is involved in, but also secondary cellular responses to the appearing mutation. The rationale for this study, that even though the mutant strains have many different mutations, mutant strains with higher protein secretion may have similar transcriptional regulatory responses caused by these different mutations. This hypothesis is confirmed by the present work, which mainly focuses on transcriptional responses to the mutations and, therefore, has less emphasis on actual mutations. From this transcriptional genome-scale analysis we can identify conserved patterns in high-protein secretion mutant strains, and reveal critical factors for efficient protein secretion in yeast.

## Results

**Phenotypic characterization.** Using ultraviolet mutagenesis and microfluidic droplet sorting, we previously isolated several different yeast strains with improved secretion of the heterologous enzyme α-amylase[16]. Here, we systematically analyzed these strains to reveal the mechanisms of efficient secretion. All the mutant strains, together with the reference strain, were grown in batch cultures in order to obtain quantitative phenotypic information (Fig. 1a). Compared with the reference strain AAC, the mutant strains produced significantly more α-amylase throughout the culture process resulting in a higher final α-amylase titer (Fig. 1b, c). An increase in the final α-amylase titer was associated with an increase in the specific α-amylase production rate, and this rate was fourfold improved for the best strain B184 compared with AAC (Table 1). Interestingly, increased α-amylase production was associated with increased specific growth rate, increased glucose uptake rate, increased ethanol production rate (Table 1, Supplementary Fig. 1a, b), and a decreased yield of biomass on glucose and an increased yield of ethanol on glucose (Supplementary Table 1). The final biomass yield of most mutant strains was, therefore, slightly lower (around 10%) (except for the strain F83 that had a 27% lower final biomass yield) compared to AAC (Supplementary Fig. 1e). It was noticed that strain M715 had the lowest specific glycerol production rate and lowest specific acetate production rate, but these rates increased again in descendants of M715 (Table 1, Supplementary Fig. 1c). The higher α-amylase titer in the medium and lower intracellular α-amylase percentage of the mutant strains showed that the secretion capacity of the mutant strain was improved (Supplementary Fig. 1d). From this phenotypic characterization of the strains it is clear that increased

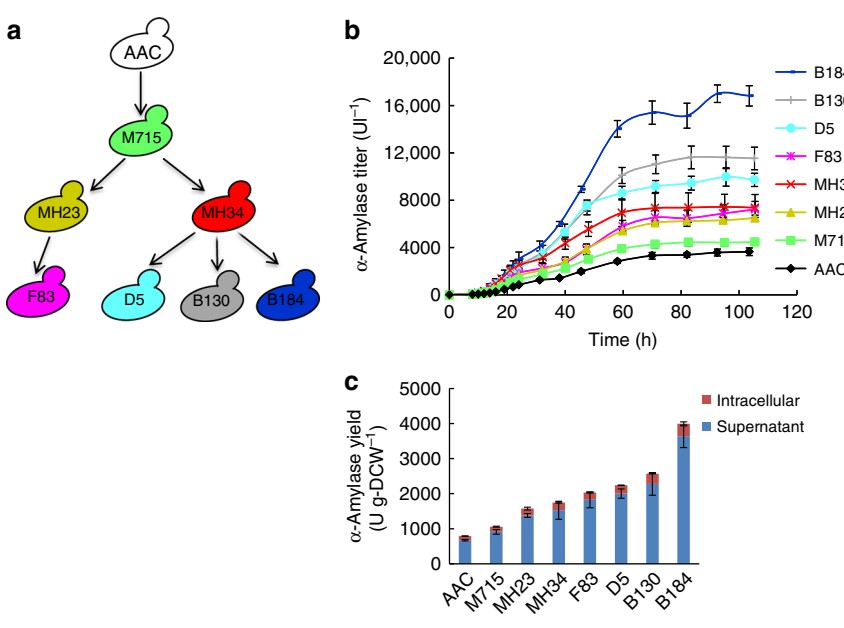

**Fig. 1** α-Amylase secretion yeast strains in batch cultures. **a** Evolutionary relationships among the strains used in this study. Strains were selected for higher α-amylase production from ultraviolet mutagenesis libraries with microfluidic screening in an earlier study[16]. **b** α-Amylase titer of strains during batch cultures. Cells were cultured with an initial OD$_{600}$ of 0.01 in SD- × SCAA medium in a bioreactor by controlled at 30 °C, 600 rpm agitation, 30 l h$^{-1}$ air flow, pH = 6. **c** The final α-amylase yield. Data shown are mean values ± standard deviations of triplicates or quadruplicates

**Table 1 Physiological parameters of the mutant strains**

| Strain | $\mu_{max}$ | $r_S$ | $r_E$ | $r_G$ | $r_A$ | $r_P$ |
|---|---|---|---|---|---|---|
| AAC | 0.276 ± 0.010 | 1.351 ± 0.024 | 0.347 ± 0.006 | 0.099 ± 0.002 | 0.040 ± 0.001 | 101.37 ± 4.90 |
| M715 | 0.309 ± 0.017 | 1.329 ± 0.249 | 0.360 ± 0.036 | 0.083 ± 0.011 | 0.033 ± 0.002 | 164.01 ± 12.61 |
| MH23 | 0.304 ± 0.007 | 1.408 ± 0.147 | 0.439 ± 0.024 | 0.091 ± 0.005 | 0.039 ± 0.003 | 223.45 ± 10.18 |
| F83 | 0.296 ± 0.005 | 1.588 ± 0.158 | 0.495 ± 0.051 | 0.117 ± 0.005 | 0.040 ± 0.005 | 264.55 ± 69.41 |
| MH34 | 0.329 ± 0.016 | 1.649 ± 0.141 | 0.460 ± 0.023 | 0.087 ± 0.006 | 0.034 ± 0.002 | 262.96 ± 35.56 |
| D5 | 0.313 ± 0.002 | 2.023 ± 0.071 | 0.604 ± 0.024 | 0.116 ± 0.007 | 0.040 ± 0.003 | 283.26 ± 15.40 |
| B130 | 0.305 ± 0.006 | 2.045 ± 0.181 | 0.677 ± 0.058 | 0.109 ± 0.015 | 0.040 ± 0.002 | 314.11 ± 73.15 |
| B184 | 0.310 ± 0.006 | 1.969 ± 0.069 | 0.610 ± 0.049 | 0.116 ± 0.013 | 0.047 ± 0.002 | 386.93 ± 49.01 |

Data shown are mean values ± standard deviations of triplicates or quadruplicates
$\mu_{max}$ maximum specific growth rate ($h^{-1}$) on glucose, $r_S$ specific glucose uptake rate (g g-DCW$^{-1}$ h$^{-1}$), $r_E$ specific ethanol production rate (g g-DCW$^{-1}$ h$^{-1}$), $r_G$ specific glycerol production rate (g g-DCW$^{-1}$ h$^{-1}$), $r_A$ specific acetate production rate (g g-DCW$^{-1}$ h$^{-1}$); $r_P$ specific α-amylase production rate (U g-DCW$^{-1}$ h$^{-1}$)

protein secretory capacity does not impose any penalty on growth as the mutant strains were growing faster, but the strains have a higher glucose uptake rate and an increasing fraction of the glucose is directed toward ethanol production. As protein synthesis and secretion are energy consuming processes, increased ratio of ATP production per total cell protein in fermentative growth[17] helps to meet the increased energy demand in strains with higher α-amylase secretion.

**Transcriptional profiling**. The relative amylase yield was calculated for the exponential phase and the end of cultivation, respectively. Mutant strains showed higher amylase yield in exponential phase compared with the reference strain AAC (Supplementary Fig. 2a, b). A transcriptional steady state was reported in exponentially growing yeast cells, and analysis of the transcriptome in the exponential phase is, therefore, reliable[18]. Therefore, cells were sampled at early exponential phase (OD$_{600}$ ≈ 1) for RNA sequencing to reveal important factors influencing protein secretion. The Spearman's correlation coefficient among samples by pairwise comparison showed strong reproducibility of biological replicates (Fig. 2a). Furthermore, this analysis showed that the strains can be classified into three groups: group 1 contained AAC and M715; group 2 contained MH23 and F83; group 3 contained MH34, D5, B130, and B184. This grouping was consistent with the evolutionary pedigree of the strains (Fig. 1a) and revealed by principal component analysis (Supplementary Fig. 3). These findings indicated that group 2 and group 3 had different evolutionary paths toward increased protein secretion. In contrast to strain M715, more genes were significantly upregulated or downregulated in higher α-amylase producing strains from group 2 and group 3 (Supplementary Fig. 4). This indicated that the significant improvement in α-amylase production obtained in strains from these two groups was associated with a global modulation of gene expression, which may be due to a requirement for adjustment of many different cellular processes to support the increased α-amylase production. Chromosome III was duplicated in MH34 and its descendants D5, B130, and B184, and in these strains, many genes located on this chromosome showed an about twofold increase at the transcription level.

To identify common expression changes in the strains in group 2 and group 3, significantly differentially expressed genes were plotted in a Venn diagram (Fig. 2b). Hereby a total of 31 commonly differentially expressed genes within all mutant strains in these two groups were identified, and the differential expression levels of these 31 genes compared with the reference strain are summarized in Fig. 2c. Several genes, including ANB1, TIR3, CYC7, DAN1, and AAC3, which are expressed under anaerobic/hypoxic conditions and/or required for anaerobic growth[19, 20], are significantly upregulated in the mutant strains.

As the dissolved oxygen levels in the medium at the time point of cell sampling for RNA extraction were around 90% for all strains (Supplementary Fig. 2c), it implied that the mutant strains exhibited anaerobic characteristics despite the aerobic environment. Most of the significantly downregulated genes were phosphate responsive genes. PHO12, PHO84, PHO89, and SPL2, and these are all related to phosphate utilization and regulation. GIT1 is involved in phospholipid metabolism and PHM6 is regulated by phosphate levels[21].

**Reporter TFs analysis and reporter gene ontology (GO) terms analysis**. To identify underlying transcriptional regulatory responses in the mutant strains, the transcriptome data were integrated with a network of transcription factors (TFs) and associated genes to identify so-called reporter TFs for mutant strains from group 2 and group 3[22, 23]. From this analysis, 308 TFs were scored. The top five reporter TFs of each strain supposedly represented an important regulatory network associated with increased protein secretion and were presented with directional significances of their target genes that were found to be upregulated or downregulated (Fig. 3a). The expression level of the reporter TFs themselves were also evaluated (Fig. 3b). In order to study whether expression level of genes on the duplicated chromosome III influenced the reporter analysis, we did a reporter TFs analysis where we removed all genes on the chr III from the analysis, and then redid the reporter TF analysis for the strains MH34, D5, B130, and B184 (Supplementary Fig. 5). Identified TFs by the reporter analysis without including genes on chr III were almost the same as the identified TFs by the reporter analysis using data for all genes (Fig. 3a), although the significances were different. Only SNF2 is not identified in the reporter analysis not including genes on chr III. This suggests that key information in global transcriptional regulatory responses, caused by mutations and chr III duplication, was not determined by expression level of genes on chr III.

Rox1p is a transcriptional regulator that represses hypoxia induced genes[20]. The reporter TFs analysis showed upregulation of genes repressed by Rox1p, which was consistent with the identification of commonly significantly upregulated genes in the mutant strains. Furthermore, expression of ROX1 gene was found to be decreased in most of the mutant strains. Gene regulated by TUP1p, an element of Tup1p–Cyc8p complex mediating repression of anaerobic genes with Rox1p[24], were also found to be upregulated in the mutant strains. Hap1p is a heme-responsive transcriptional activator and transcriptional repressor under aerobic and anaerobic conditions, respectively[25]. HAP2, HAP3, HAP4, and HAP5 encode proteins to form the Hap2p/3p/4p/5p CCAAT-binding complex, which is a heme-activated and glucose-repressed transcriptional activator and global regulator of respiratory gene expression[26]. Genes regulated by Hap1p and

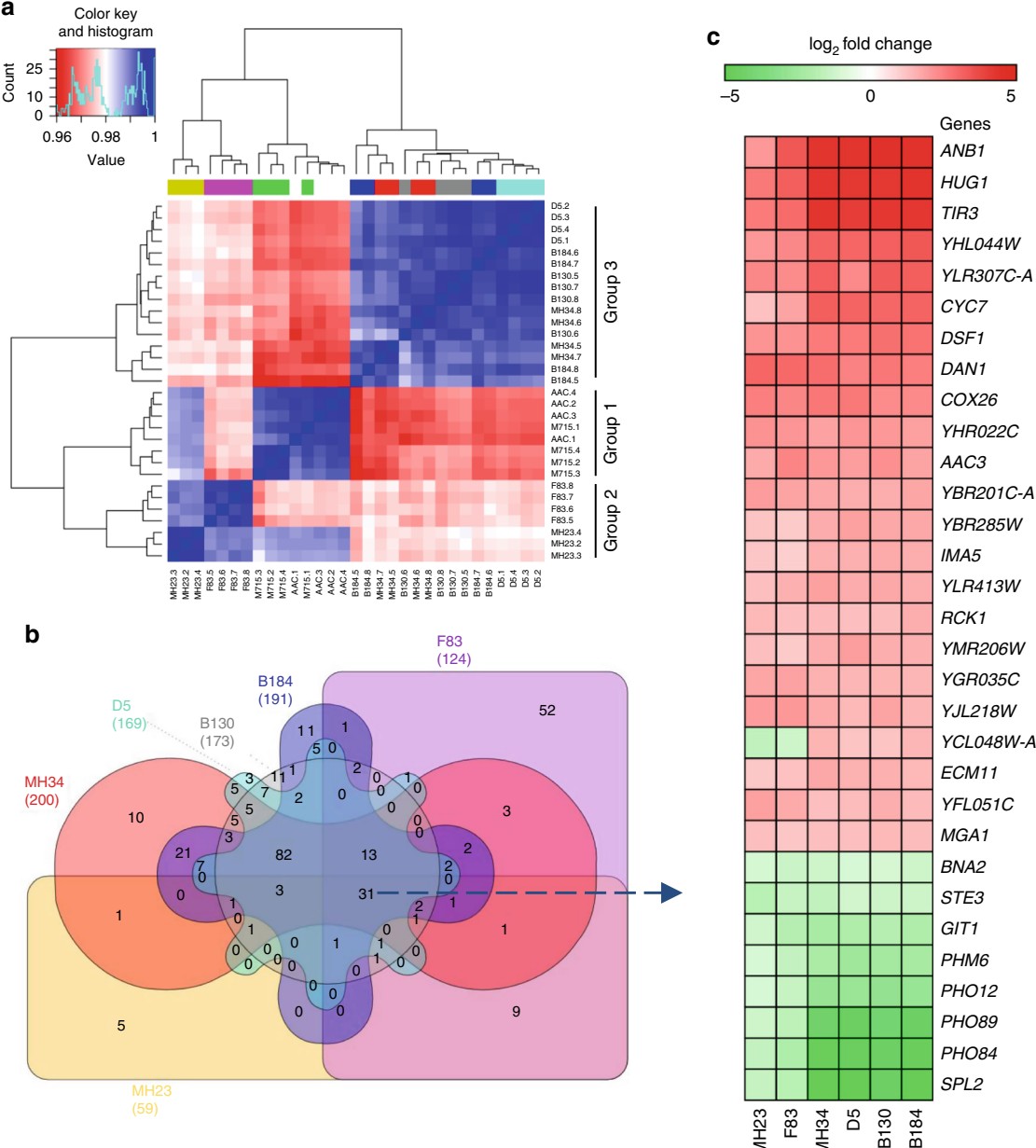

**Fig. 2** Overview of transcriptome data. **a** Heatmap of Spearman's correlation coefficient for the similarities in the expression profiles of exponential phase between the different samples by pairwise comparison. **b** Common very significantly differentially expressed genes ($P$-adj < 0.05 (Benjamini–Hochberg method) and abs ($\log_2$ Fold change) > 1) in the mutant strains compared with the reference strain AAC. The number of very significantly differentially expressed genes in each strain is specified under the strain name. **c** Expression levels of common significantly differentially expressed genes in mutant strains

the Hap2p/3p/4p/5p CCAAT-binding complex were found to be downregulated in the mutant strains. All these results indicated that the mutant strains were expressing genes as if they were in hypoxia, which may facilitate efficient protein secretion.

Besides anaerobic metabolism, other types of regulation were identified, including nutrient signaling, nucleotide synthesis, and phosphate metabolism, cell cycle, etc. by the reporter TFs analysis (Fig. 3a). *MSS11* and *TEC1* are involved in response to starvation and responsible for nutrient regulation[27], and both represent reporter TFs for upregulated genes. As production of heterologous protein competes with intracellular resources that could be used for cell growth, slightly lower final biomass yields were found for the efficient protein secretion strains (Supplementary Fig. 1e). Actually, the α-amylase produced by the best production

strain B184 accounted for about 13% of total cellular protein produced (Supplementary Fig. 2d). Strong competition for resources may result in induction of a partial nutrient starvation response, hence activated nutrient related regulators like Mss11p and Tec1p. Bas1p is responsible for inducing expression of genes involved in nucleotide synthesis and phosphate consumption by interaction with Pho2p[28]. Downregulation of genes regulated by Bas1p in the mutant strains hinted that protein secretion may benefit from this kind of response. The first two steps of de novo synthesis of pyrimidine are catalyzed by Ura2p, which is feedback inhibited by the level of UTP. The *URA2* gene was found to be downregulation ($P$-adj < 0.05, Benjamini-Hochberg method) in our mutant strains. *FUR1*, which encodes enzyme for conversion of uracil into UMP in the salvage pathway, was found to be

upregulation in the mutant strains, which indicates that nucleotide biosynthesis may shift to use of the salvage pathway. The reporter TFs analysis also revealed that genes regulated by Mbp1p and Swi4p were upregulated. Both Mbp1p and Swi4p can interact with Swi6p to form a MBF (Mbp1p/Swi6p-dependent cell cycle box Binding Factor) complex and a SBF (Swi4p/Swi6p-dependent cell cycle box Binding Factor) complex, respectively,

which regulate gene expression during the G1/S transition of the cell cycle[29].

To validate if the identified reporter TFs represent key points in regulation of protein secretion, some of the TFs were selected for evaluation. Previously engineering of TFs through gene deletion or over-expression has shown to impact protein secretion, and our TF reporter analysis of the mutant strains is consistent with

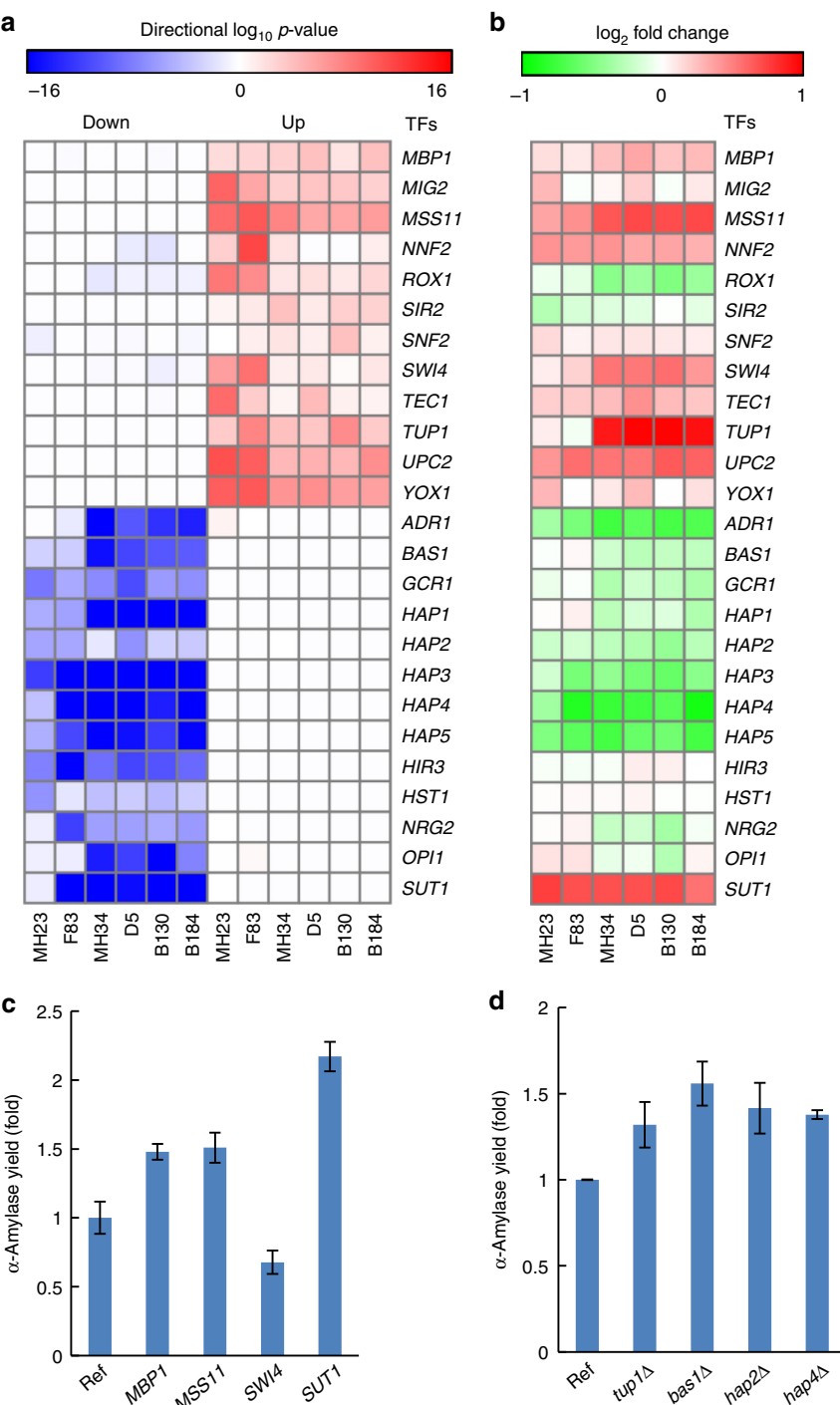

Fig. 3 Reporter transcription factors (TFs) analysis revealed important transcriptional network responses in mutant strains. TFs were scored by the modulation in expression level of genes that controlled by TFs. **a** The top five scored reporter TFs for each strain in distinct-directional up class (red) and distinct-directional down class (blue) are chosen and presented by their significance. **b** Expression levels of reported TFs in the mutant strains compared with the reference strain AAC. Enhanced production of α-amylase by overexpression of TFs **c** or deletion of TFs **d**, data shown are mean values ± standard deviations of duplicates

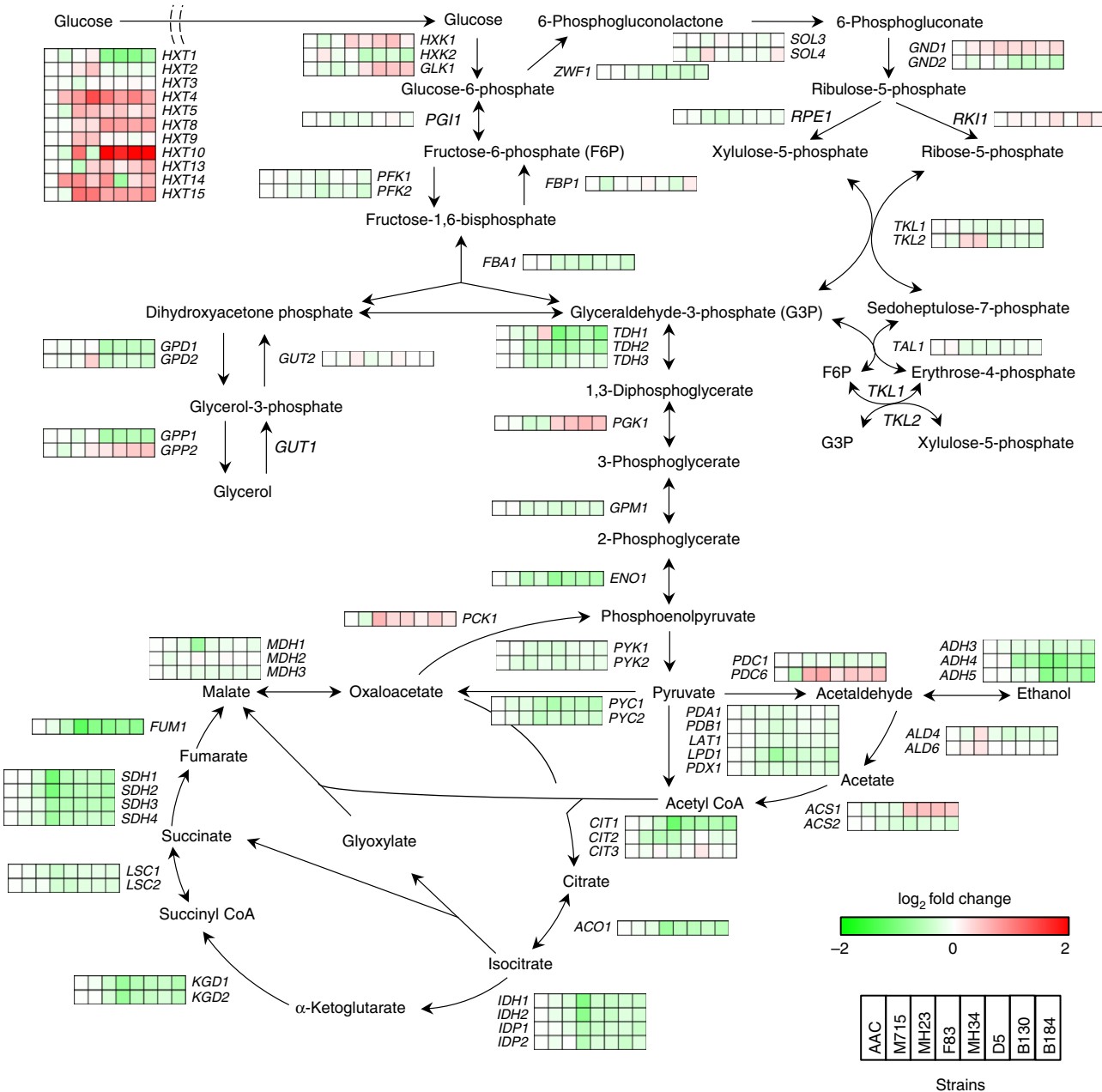

**Fig. 4** Transcriptional changes of genes involved in carbohydrate metabolism. Most genes of the central carbon metabolism were downregulated and genes encoding high-affinity hexose transporters were upregulated in the mutant strains

these studies. Thus, the TF reporter analysis indicates that deletion of *ROX1* or upregulation of *UPC2* should improve protein production, which is indeed the case[30], and that deletion of *HAP1* should improve protein production[31]. To evaluate if some of the additional reporter TFs identified in this study have an impact on protein secretion, we deleted or over-expressed some of the respective genes according to their altered expression level in the mutant strains. Hence, *MBP1*, *MSS11*, *SWI4*, and *SUT1* were over-expressed and *BAS1*, *HAP2*, and *HAP4* were deleted in strain AAC (Fig. 3c, d). *TUP1* is located on chromosome III and had high expression level in the mutant strains, due to duplication of the entire chromosome III. As mentioned above Tup1p interacts with Rox1p, and as deletion of *ROX1* has been found to have a positive impact on protein secretion we also deleted *TUP1* (Fig. 3d). Except for over-expression of *SWI4*, all the engineering strategies resulted in

increased protein secretion, and this shows that efficient protein secretion can be achieved by global tuning of gene expression by engineering of TF expression. Furthermore, these findings provide a validation that computationally identified reporter TFs for mutant strains can be used as targets for improving protein secretion.

We also calculated reporter GO terms for the mutant strains from group 2 and group 3. Hereby we found that genes associated with GO terms related to mitochondrial function, generation of precursor metabolites and energy, cellular respiration, cellular amino-acid metabolic process, etc. were downregulated. In contrast, genes associated with GO terms related to ribosome function, cytoplasmic translation, regulation of organelle organization, Golgi vesicle transport, protein lipidation, lipid metabolic process etc. were upregulated (Supplementary Fig. 6). These results were consistent with the phenotypic analysis that showed

that the mutant strains exhibited reduced respiration and increased protein synthesis and protein trafficking. Furthermore, changes of lipid metabolism in the mutant strains may contribute to increased protein secretion[32, 33].

We also experimentally verified some of the nonsense mutations in genes associated with identified GO terms. Thus, *PGM2*, encoding phosphoglucomutase involved in glycogen biosynthesis, and *PXA1*, encoding one of the subunits for a fatty-acid transporter to the peroxisomes, carry nonsense mutations and are hence non-functional. We, therefore, evaluated the impact of deleting these genes on protein secretion in the reference strain AACK and found that these deletions results in 30–45% improvement in protein production (Supplementary Fig. 7), which hereby supported our GO term analysis.

**Transcription analysis of genes in carbohydrate metabolism.** As macroscopic fluxes such as glucose uptake rate and ethanol production rate were changed in the mutant strains, and at the same time GO terms associated with cellular respiration and generation of precursor metabolites and energy metabolism were found to be downregulated, we investigated the transcriptional levels of genes related to carbohydrate metabolism in the mutant strains. The expression levels of genes were presented on a map of the central carbon metabolism (Fig. 4). *HXT1*, encoding a low-affinity glucose transporter[34], was found to be downregulated in mutant strains from group 3. In contrast, genes encoding high-affinity hexose transporters were upregulated in almost all mutant strains. This altered transcriptional level of hexose transporter genes was inconsistent with the high glucose concentration in the medium, and, therefore, indicated abnormal glucose sensing in the mutant strains, but still resulted in an increased glucose uptake rate (Table 1). By GO slim mapper analysis of mutations and differential expressed genes in the mutant strains[16] (Supplementary Data 1), a mutant gene *SNF3* was found to be associated with the GO term carbohydrate transport. As SNF3p is a plasma membrane glucose sensor involved in regulation of expression of hexose transporters[35], abnormal glucose sensing and altered transcriptional level of hexose transporter genes were likely affected by the mutation in *SNF3*. Most genes of the central carbon metabolism were downregulated in the mutant strains. For the TCA cycle, this was consistent with the reduced respiration and increased ethanol production in these strains, whereas decreased expression of glycolytic genes seemed inconsistent with the increased glucose uptake rate. However, glycolytic flux is well known to not primarily be controlled at the transcriptional level, contrary to the TCA cycle and respiration[35].

Amino acids are building blocks for protein synthesis. Many genes involved in amino acid biosynthesis pathway were downregulated in the mutant strains (Supplementary Fig. 8). Genes *CHA1*, *HIS4*, *ILV6*, and *THR4* were found to be upregulated in strains MH34, D5, B130, and B184, but this can be ascribed to the location of these genes on the duplicated chromosome III. *GDH3*, *GLN1*, and *GLT1* involved in glutamate and glutamine biosynthesis were also upregulated[36]. Both glutamate and glutamine serve as amino donors. Furthermore, genes *AAT1*, *ALT1*, and *BAT1*, which encode transaminases for reversible conversion between glutamate and other amino acids[37–39], were upregulated. These results indicated that glutamate and glutamine play important roles in protein synthesis. Conversion of amino acids through transaminases to balance the intracellular amino acids pool is critical for efficient protein production, in particular when amino acids are supplied via the medium as in our case. When comparing the amino acid composition of α-amylase with that of yeast cell proteins, it was found that there is a 9.3-fold higher requirement for cysteine for production of the same amount of α-

amylase compared with the average biosynthesis of the same amount of yeast cell proteins (Supplementary Table 2). In fact, the requirement of cysteine in the mutant strains increased by 25–85% compared with the reference strain AAC (Supplementary Table 3). However, genes of the cysteine biosynthetic pathway still did not show increased expression (Supplementary Fig. 8). Yet, from analysis of genes involved in amino acid transport, we found that *YCT1* and *ERS1* responsible for cysteine transport were significantly upregulated[40, 41] (Supplementary Fig. 9), and this can help the mutant strains to meet the increased demand for cysteine when α-amylase production is increased (Supplementary Table 3).

**Analysis and engineering of thiamine biosynthesis.** In our previous study[16], the *Rhizopus oryzae* glucan-1,4-α-glucosidase was shown also to have increased secretion by the mutant strains, and as for α-amylase strain B184 was by far the best in terms of protein secretion. Here, we, therefore, tested this strain for its ability to produce two other proteins, human serum albumin, and *Trichoderma reesei* endo-1,4-beta-xylanase II. Compared with the reference strain CEN.PK 530.1C, both proteins showed higher protein yield in B184 (Supplementary Fig. 10). These results suggested our findings were not limited to a specific protein and may be generally applicable. We, therefore, performed a more detailed analysis of the transcriptome of the best strain B184 in terms of protein secretion.

Significantly upregulated and downregulated genes in this strain were analyzed by GO bioprocess enrichment (Fig. 5a and Supplementary Fig. 11), and this analysis pointed to genes involved in thiamine-biosynthesis having significantly increased expression (Fig. 5a). The GO term associated with thiamine-containing compound metabolism includes thiamine biosynthetic process, thiamine-containing compound biosynthetic process and thiamine metabolic process. By overlaying gene expression for the strains on a pathway map for thiamine biosynthesis it was further clear that the expression levels of genes related to this pathway were upregulated in strain B184 (Fig. 5b). It was noticed that these genes were also upregulated in strains F83, D5, and B130. All these four strains were derived from the second round of screening and showed higher protein secretion level compared with their ancestral strains. Based on this information, we speculated that thiamine may be required for efficient protein secretion. We, therefore, performed experiments with additional thiamine supplemented to the medium, but from these experiments it was found that the reference strain AAC showed no difference in α-amylase secretion and the best mutant strain B184 even exhibited a small decrease in α-amylase secretion upon additional thiamine supplementation (Fig. 5c). We then further investigated the transcriptional control of thiamine biosynthetic genes. It has been reported that expression of thiamine biosynthetic genes is controlled by Thi2p together with Thi3p and Pdc2p, and transcription of *THI2* is regulated negatively by the intracellular thiamine level[42]. As the expression levels of *THI2* and *THI3* in the mutant strains were upregulated, this implied that the cells secreting efficiently α-amylase may be in a low cellular thiamine status. Yet the low thiamine status triggered thiamine response mechanism, which attempted to counter-balance thiamine level by elevated the transcription of thiamine biosynthetic genes. Hence, *THI2*, *THI3*, and *THI4* were deleted in strains AAC and B184 to reduce thiamine biosynthesis, and the single gene deletion resulted in increased protein secretion in both strain backgrounds (Fig. 5d). Several thiamine containing proteins (thiamin diphosphate-dependent enzymes), encoded by *ARO10*, *ILV2*, *KGD1*, and *THI3*, are involved in amino-acid metabolism (Fig. 5e)[42, 43], and their expression levels were lower

in the mutant strains. Hence, a decrease thiamine concentration may additionally reduce the activity of these enzymes, suggesting that attenuating activity of these pathways may benefit protein secretion. We, therefore, deleted these genes in AAC, and deletion of each of these genes resulted in increased protein secretion (Fig. 5f).

**Reduced ER stress analysis.** Heterologous protein production often brings a protein folding burden to the cell and, therefore, causes oxidative stress in the endoplasmic reticulum (ER)[44]. The unfolded protein response (UPR) pathway is activated by arising oxidative stress in the ER to assist in reducing cellular stress. Hac1p is a key UPR-induced TF for transcriptional activation of

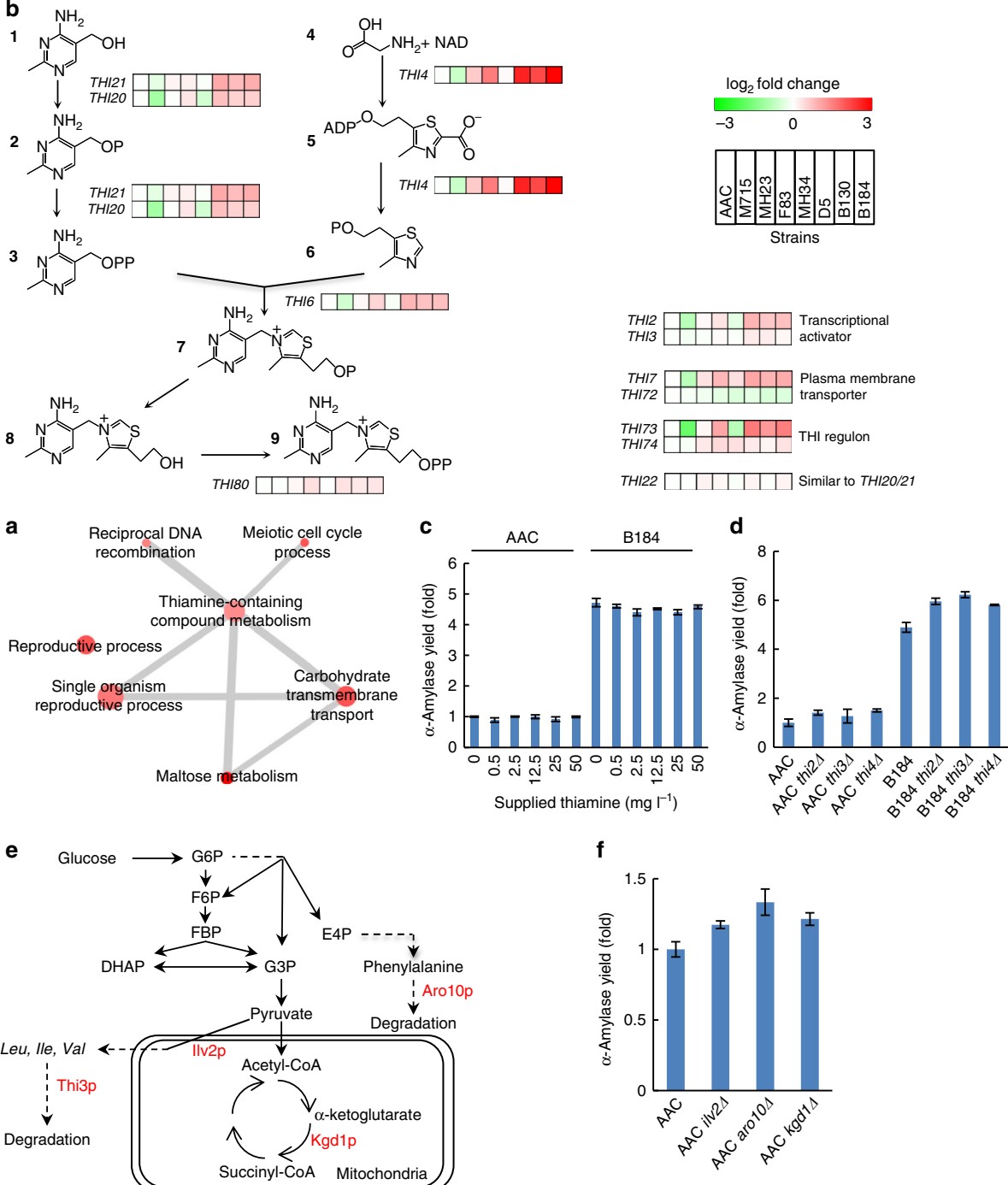

**Fig. 5** Low thiamine status is favorable for α-amylase production. **a** GO bioprocess enrichment of most upregulation genes (P-adj < 0.05 (Benjamini–Hochberg method) and log₂ fold change > 0.5) in strain B184. **b** Transcriptional levels of genes related to thiamine biosynthesis. 1: hydroxymethylpyrimidine; 2: hydroxymethylpyrimidine phosphate; 3: 2-methyl-4-amino-5-hydroxymethylpyrimidine diphosphate; 4: L-glycine; 5: adenylated thiazole; 6: 4-methyl-5-(β-hydroxyethyl)thiazole phosphate; 7: thiamine phosphate; 8: thiamine; 9: thiamine diphosphate. **c** α-Amylase production in the SD-2 × SCAA medium supplied with different amount of thiamine. **d** Deletion of THI2, THI3, and THI4 enhances α-amylase production. **e** Thiamine diphosphate-dependent enzymes involved in amino-acid metabolism. DHAP dihydroxyacetone phosphate; E4P erythrose-4-phosphate; F6P, fructose-6-phosphate; FBP, fructose-1,6-bisphosphate; G6P, glucose-6-phosphate; G3P, glyceraldehyde-3-phosphate. **f** Deletion of ILV2, ARO10, and KGD1 enhances α-amylase production. Data shown in **c**, **d**, **f** are mean values ± standard deviations of duplicates

ER chaperone encoding genes, including KAR2, ERO1, etc[45]. Overexpression of HAC1 has been reported to enhance protein secretion[46]. Mutant strains MH23 and F83 from group 2 were found to have increased expression levels of HAC1 and ERO1. In contrast to group 2, mutant strains MH34, D5, B130, and B184 from group 3 were found to have HAC1, ERO1, and KAR2 downregulated (Supplementary Fig. 12a). However, it was noticed that both PDI1 and EMC1 responsible for efficient folding of proteins in the ER were upregulated in strains MH34, D5, B130, and B184. The upregulation of PDI1 and EMC1 was due to their location on the duplicated chromosome III. Several other genes involved in protein folding, EMC2-6, were found to be upregulated, and the expression extent was more significant in strains from group 3. These results suggested that strains from group 2 and group 3 used different mechanisms to release oxidative stress from heterologous protein production. Group 2 mainly relied on the UPR pathway, whereas group 3 relied on direct up-regulation of key chaperones in the ER enabling more efficient protein folding.

In order to evaluate whether the altered expression of chaperones in the ER affected the oxidative stress, we measured the reactive oxygen species (ROS) level in the strains. Compared with a non-producing control strain, all α-amylase production strains had higher ROS level, indicating increased ER stress due to heterologous protein production (Supplementary Fig. 12b). Within these α-amylase production strains, the wild-type strain AAC showed the highest ROS level. This indicated that a reduction of oxidative stress can be achieved by both activation of UPR and enhancement of the protein folding capacity. Accumulated ROS was also observed visually using fluorescence microscopy (Supplementary Fig. 12d). Considering that the mutant strains produce more α-amylase, the data on ROS accumulation point to that the ROS generation associated with heterologous protein production, i.e., ROS per α-amylase yield unit, was reduced in the mutant strains (Supplementary Fig. 12c). A reduced oxidative stress in mutant strains was also supported by the decreased expression levels of oxidative stress response genes (Supplementary Fig. 12a).

The mutant strains carry mutations in several genes related to ER stress or organelle transport, such as EMC1, ERV29, USO1, VPS10, SNC2, which is consistent with our findings from the transcriptome analysis. In our previous study[16] we confirmed protein secretion was associated with ERV29 and SNC2; and deletion of SNC2 resulted in improved protein production. To further evaluate whether there is a general impact of modulating ER stress or organelle transport we further evaluated EMC1, USO1, and VPS10. These genes were deleted in AACK and deletion of EMC1 and VPS10 was found to improve protein production by 10–40%, whereas deletion of USO1 did not impact protein production (Supplementary Fig. 7). The mutant strains also have gene duplications in genes involved in ER stress, e.g., PDI1, and we, therefore, over-expressed this in AACK and found that this results in improvement of protein production.

The reduced oxidative stress could also be related to an improved supply of NADPH. Increased supply of NADPH via the pentose phosphate pathway (PPP) was shown beneficial for higher protein production in Chinese Hamster Ovary (CHO) cells and Pichia pastoris[47, 48]. Although no increased expression was seen for PPP associated genes, we found reduced biomass yields of the mutant strains. As many biosynthesis pathways require NADPH as reducing equivalents, a decreased biomass yield will allow the cells to save more NADPH for maintaining redox balance (protection against oxidative stress). The mutant strains, therefore, seemed to meet the demands for NADPH associated with increased amylase production by redistribution of resources. This is supported by the total protein content in the mutant strains. As amylase content increased, the total protein (yeast cell protein and amylase) did not increase in the mutant strains compared with the reference strain AAC (Supplementary Fig. 13).

## Discussion

Here, we studied some of the underlying mechanisms of efficient protein secretion through comparative systems biology analysis of efficient α-amylase secretion mutant strains and a reference strain. From genome-wide transcription analysis we found that the majority of genes related to glycolysis and TCA cycle were downregulated in the mutant strains, but still the final biomass yield was only slightly decreased and the maximum specific growth rate was even increased. Previously, it was found that glycolytic enzymes represent a large proportion (30–60%) of the soluble proteins in the cell[49], and a recent study reported that there are strong redundancies in glycolytic enzymes so that reducing the level of several glycolytic proteins only has a minor impact on yeast growth[50]. Downregulation of glycolytic genes may, therefore, have occurred in order to allocate more proteome mass and protein synthesis capacity for heterologous protein production. Attenuation of phosphoglucomutase activity (encoded by PGM2) through inactivation of PGM2 may have further contributed to this effect, as confirmed by a 40% increase in protein production by deletion of this single gene (Supplementary Fig. 7). Furthermore, allocation of more resources to protein secretion resulted in a reduction of cellular stress, allowing the cells to grow slightly faster. Faster growth is associated with increased ethanol production, and, therefore, a slightly lower biomass yield. It is also interesting that even though the cells have an increased amino acid demand for production of α-amylase, genes associated with amino acid biosynthesis were downregulated in the mutant strains, but this was compensated by increased expression of amino acid transporters. Thus, the mutant strains seem to shift from amino acid biosynthesis to increased amino acid uptake. Yet genes involved in glutamate/ glutamine biosynthesis and amino-group transfer (transaminases) were upregulated, which is necessary to balance the requirement for amino acids for cell growth and α-amylase production to the provision of amino acids from the extracellular medium. In addition to serving as building blocks of proteins, glutamate and glutamine play important roles in many other processes (e.g., oxidative stress responses, nucleotide metabolism), and increased expression of genes involved in glutamate/glutamine biosynthesis may, therefore, also be due to demands from other processes. The gene ARO10 encodes a protein (decarboxylase) involved in amino acid degradation. Improvement of amylase production by deletion of ARO10 also highlights the importance of maintaining amino acid pool by reduced amino acid degradation for efficient protein production.

It is noteworthy that many identified reporter TFs were related to anaerobic conditions. Hypoxic genes controlled by reporter TFs were upregulated in the mutant strains, whereas respiratory genes were down-regulated. This was consistent with reporter GO terms analysis, which showed that cellular respiration and mitochondrial function were downregulated. A previous study showed that deletion of ROX1, which results in de-repressed expression of anaerobic genes at aerobic conditions, caused increased heterologous protein secretion[30]. Genes controlled by Upc2p and UPC2 itself were found upregulated in the rox1Δ strain, which was also found in our mutant strains. Increased protein secretion was also shown by overexpression of the UPC2-1 allele, which constitutively activates ergosterol biosynthesis genes. These results indicated that anaerobic characteristics of the mutant strains were favorable for protein

secretion. This was supported by improved amylase production in a *KGD1* deletion strain, which has a decreased respiratory growth. Elevated oxidative stress due to an increased burden from protein folding hinders protein production, and our analysis shows that this can be overcome either by increased expression of protein folding chaperones or activation of the UPR[51]. Our results emphasize the importance of reducing oxidative stress associated with protein production regardless of the pathway used for this. Again, herein reducing oxidative stress in the mutant strains with evidences at transcriptional levels and by experimental validation (ROS level) supported the importance of UPR and HSR found in our previous study[16]. Hence, mutant genes and duplicated genes related to ER or organelle transport (such as *EMC1, ERV29, PDI1, USO1, VPS10, SNC2*) in the mutant strains may contribute to reduced oxidative stress and facilitate protein secretion. This was confirmed by deletion or over-expression of these genes in the reference strain (Supplementary Fig. 7). It has also been reported that a mutant *EMC1* causes induction of the UPR[8], and this can partly explain our finding of a UPR in strain MH23 and F83, both of which had a mutant *EMC1*[16]. Our results also supported the findings from another previous study[52], in which Wentz et al.[52] used a rapid flow cytometric sorting method to screen yeast complementary DNA overexpression libraries for increasing protein surface display. *ERO1* was one of hits in the study by Wentz et al. and overexpression of *ERO1* increased protein secretion through assisting protein folding. As improved supply of NADPH was beneficial for protein production by protection against oxidative stress[47, 48], although no increased expression changes were seen in PPP in our mutant strains, it would be interesting to metabolic engineer the PPP in our mutant strains for improved protein production in future studies. Besides alteration of intracellular processes by changes in gene expression, cells can also alter processes by regulating the activity of enzymes[53, 54] and our results showed that several pathways catalyzed by thiamine-dependent enzymes were affected by a low thiamine status in the mutant strains[55], which may provide another solution to tune the activity of metabolic pathways. Although valuable findings were found based on transcriptional data from the exponential phase, it may be worthwhile to investigate the transcriptome from other growth phases as follow-up studies.

We believe our study is the first systems level analysis of strains with various levels of protein secretion. As the strains used in our analysis possess a large number of different genomic mutations causing changes in many different intracellular processes it would be difficult to identify the effects of these mutations. However, from our systems level analysis we could identify common regulation patterns and hereby we could specify some general rules for efficient protein secretion. We confirmed several of these findings through inverse metabolic engineering where we altered the expression of identified reporter TFs. As *S. cerevisiae* is both an industrially important cell factory for recombinant protein production and a key eukaryal model organism, our findings can most likely be used as guidelines for design of other cell factories for efficient protein secretion, e.g., filamentous fungi used for production of industrial enzymes and CHO cells used for production of pharmaceutical proteins. Furthermore, our findings may also contribute gaining improved insight into the mechanism of the human protein secretory pathway. Dysfunction of this pathway is associated with many different diseases[56], and it may even be possible to use our strains as a platform for screening for drugs that can cure diseases related to protein secretion[57, 58].

## Methods

**Strains and plasmids**. All strains and plasmids used in this study are listed in Supplementary Table 4. All primers used in this study are listed in Supplementary

Table 5. Plasmids for gene overexpression were constructed by insertion of the gene fragment, which was amplified from the yeast genome by corresponding primer pairs and digested with restriction enzymes, to the expression vector pSPGM1 (Supplementary Fig. 14). Synthesized human serum albumin (HSA) gene with alpha factor leader and *Trichoderma reesei* endo-1,4-beta-xylanase II gene with alpha factor leader were cloned in plasmid CPOTud, resulting in plasmids pCP-AHSA and pCP-AXYN2, respectively (Supplementary Fig. 10a, b). Single gene deletion and promoter replacement were performed using amdS as a selection marker, and transformants were selected on SM-Ac plate[59]. The deletion cassette was constructed by amplification of the amdS marker from the plasmid pUG-amdSYM with primer pairs containing regions homologous to the target gene. The standard LiAc/SS DNA/PEG method was used for yeast transformation[60].

**Media and culture conditions**. Yeast strains were grown in YPD medium, YPE medium or SD-URA medium at 30 °C according to the auxotrophy of the cells, phenotypes of which were described in Supplementary Table 4[61]. Single gene deletion strains were selected on SM-Ac plate[59]. For α-amylase production in tubes or shake flasks, yeast strains were cultured for 96 h at 200 rpm, 30 °C in the SD-2 × SCAA medium[61] containing 20 g l$^{-1}$ glucose, 6.9 g l$^{-1}$ yeast nitrogen base without amino acids, 190 mg l$^{-1}$ Arg, 400 mg l$^{-1}$ Asp, 1260 mg l$^{-1}$ Glu, 130 mg l$^{-1}$ Gly, 140 mg l$^{-1}$ His, 290 mg l$^{-1}$ Ile, 400 mg l$^{-1}$ Leu, 440 mg l$^{-1}$ Lys, 108 mg l$^{-1}$ Met, 200 mg l$^{-1}$ Phe, 220 mg l$^{-1}$ Thr, 40 mg l$^{-1}$ Trp, 52 mg l$^{-1}$ Tyr, 380 mg l$^{-1}$ Val, 1 g l$^{-1}$ BSA, 5.4 g l$^{-1}$ Na$_2$HPO$_4$, and 8.56 g l$^{-1}$ NaH$_2$PO$_4$·H$_2$O (pH = 6.0 by NaOH). For bioreactor batch cultures, 5.4 g l$^{-1}$ Na$_2$HPO$_4$ and 8.56 g l$^{-1}$ NaH$_2$PO$_4$·H$_2$O in the SD-2 × SCAA medium were replaced by 2 g l$^{-1}$ KH$_2$PO$_4$ (pH = 6.0 by NaOH). Seed cultures were used to inoculated 500 ml SD-2 × SCAA medium in 1 l bioreactor vessels (DasGip, Germany) with an initial OD$_{600}$ of 0.01. The bioreactor system was run at 30 °C, 600 rpm agitation, 30 l h$^{-1}$ air flow, pH = 6 (controlled by NaOH). Biological quadruplicate (in some cases triplicate) experiments were conducted for each strain.

**Analytical procedures**. For dry cell weight (DCW) determination, yeast cells in 5 ml culture were harvested by a 0.45 μm nitrocellulose filter and washed with distilled water. The DCW was measured until the cells were dried to a constant by 15 min microwave heating and then 3 days in a silica gel drier. The concentration of metabolites (glucose, ethanol, glycerol, etc.) in the culture was measured by loading the supernatant to a HPX-87H column (Bio-Rad, USA) on a Dionex Ultimate 3000 HPLC system (Dionex Softron GmbH, Germany). The HPLC system was run at 45 °C with 5 mM H$_2$SO$_4$ as mobile phase at a flow rate of 0.6 ml min$^{-1}$.

**Protein quantification**. The α-amylase activity was measured using the α-amylase assay kit (Cat No. K-CERA; Megazyme, Ireland) with a commercial α-amylase from *Aspergillus oryzae* (Cat No. 86250; Sigma, USA) as a standard. The weight of α-amylase can be calculated with 69.6 U mg$^{-1}$ as protein conversion coefficient[62]. Five-hundred microliters cell cultures were centrifuged at 16,000×*g* for separation of supernatant and cell pellet. Then the culture supernatant was used for determination of α-amylase activity. For intracellular α-amylase measurements, the cell pellet was washed with distilled water and resuspended in 500 μl phosphate-buffered saline (PBS) with 5 μl halt protease inhibitor cocktail (Cat No. 87786; Thermo Fisher, USA). The cell suspension was added to lysing matrix tube and cell lysis was performed using a FastPrep-24 tissue and cell homogenizer (MP Bio-medicals, USA) by two 60 s cycles at a speed of 6.5 m s$^{-1}$ (samples were put on ice for 5 min between the two cycles). Cell debris was removed by centrifugation and the supernatant fraction was used for α-amylase quantification. Cells expressing HSA or XYN were cultured in SD-2 × SCAA medium without BSA. After cultivation, the supernatant was collected by centrifugation at 16,000×*g* for sodium dodecyl sulfate polyacrylamide gel electrophoresis (SDS/PAGE) analysis[16]. SDS/PAGE gel was analyzed by the software ImageJ and concentration of HSA and XYN was estimated[63].

**Transcriptome profiling**. Cell samples for RNA-seq were taken at the early exponential phase (OD$_{600}$ ≈ 1) and stored at −80 °C until processing[64]. RNA was extracted using the RNeasy Mini kit (Cat No. 74104; Qiagen, Germany) and prepared for sequencing using the Illumina TruSeq samples preparation kit v2, with poly-A enrichment. The fragments were clustered on cBot and sequenced on a single lane on a HiSeq 2500 with paired ends (2 × 125 bp), according to the manufacturer's instructions. The number of read pairs obtained for each sample ranged from 5.9–10.1 million. The raw data can be downloaded from the European Nucleotide Archive with access number ERP019558. The raw reads from each samples were mapped to the CEN.PK 113-7D reference genome (http://cenpk.tudelft.nl) using TopHat (v 2.0.12)[65], with 84.0—87.6% of the reads successfully mapped. Cufflinks version 2.2.1[66] was used to calculate FPKM values, and the feature Counts module of the Subread package (v 1.4.6)[67] was used to determine raw read counts. Analysis of differential expression was performed using DESeq[68]. Reporter GO terms and reporter TFs analysis were performed using the Platform for Integrative Analysis of Omics (PIANO) R package[23] with GO terms information from the Saccharomyces Genome Database (http://www.yeastgenome.org) and information on regulatory interactions between TFs and genes from Yeastract (http://www.yeastract.com). Differential expression level of genes

(log$_2$FoldChange) and corresponding significant levels (adjusted $p$-values, calculated by the Benjamini–Hochberg method) were used as input. The reporter TFs analysis was used to calculate the significance for expression change of gene sets controlled by TFs, which were scored by the modulation in expression level of genes that are controlled by a given TFs[22]. The top five TFs, in distinct-directional up class and in distinct-directional down class, were chosen and presented in significances. The reporter GO terms analysis was similar to reporter TFs analysis, but gene sets were classified by GO terms. All GO terms were scored by modulation in expression level of genes within the same GO term. The Venn diagram was generated by using InteractiVenn[69]. GO slim mapper analysis was performed by using the SGD online tool according to the instruction (http://www.yeastgenome.org/cgi-bin/GO/goSlimMapper.pl).

**ROS measurements**. The yeast cells were grown in SD-2 × SCAA medium and harvested at an OD$_{600}$ of 1–2 by centrifugation for ROS measurements. The cell pellets were washed with PBS twice, 50 mM sodium citrate buffer (SCB) once and resuspended in SCB to an OD$_{600}$ of 1. One milliliter cell suspension was combined with 1 μl 50 mM dihydrorhodamine 123 (DHR123) and incubated in the dark at room temperature for 30 min. Then, the cell pellet was washed twice with SCB and resuspended in 1 ml SCB for fluorescence intensity measurement and imaging. The fluorescence intensity was measured in a 96-well plate using a FLUOstar Omega microplate reader (BMG Labtech, Germany) at excitation wavelength 485 nm and emission wavelength 520 nm. Cells were imaged using a fluorescence microscope (Leica DMI4000B, Germany) with DIC and YFP filters.

**Data availability**. The RNA-seq raw data of the mutant strains and the reference strain can be downloaded from the European Nucleotide Archive with the access number ERP019558. The data that support the findings of this study are available from the corresponding author upon reasonable request.

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

## Acknowledgements

This work was funded by the Novo Nordisk Foundation, Vetenskapsrådet, FORMAS, and Knut and Alice Wallenberg Foundation. We would like to acknowledge the Science for Life Laboratory, the National Genomics Infrastructure (NGI), and Uppmax for providing assistance in massive parallel sequencing and computational infrastructure. We also thank Verena Siewers, Yongjin Zhou, Guokun Wang, Jiufu Qin, Mark Bis-schops, Yun Chen, and José L. Martínez for useful discussions and comments.

## Author contributions

M.H. and J.N. conceived and designed the study. M.H. and J.B. performed experiments. M.H. and B.M.H. performed bioinformatics analysis. M.H., D.P., and J.N. analyzed the data and wrote the paper.

## Additional information

**Competing interests:** The authors declare no competing financial interests.

