## [Peer Review File · Nature Communications]

Reviewers' comments:

Reviewer #1 (Remarks to the Author):

Huang and colleagues used genome-wide transcriptional analysis to investigate *Saccharomyces cerevisiae* mutants derived from the "strain AAC" and having increased efficiency in secretion of recombinant proteins. Results were validated using inverse engineering and physiological characterization. The paper is clear and well-written, some typos are present in the Methods section and a careful revision is suggested. The mutant strains investigated here are described in a paper previously published by the same authors. In this previous paper yeast mutants were analyzed using high throughput sequencing and a number of mutations on each strain were identified. However, there is very minor connection between the results reported in these two papers, while a more thorough analysis and a more clear comparison could provide a better comprehension of the results obtained. Additionally, in the present paper authors investigated the transcriptional regulatory responses in the mutant strains and integrated the transcriptome data with a network of transcription factors and associated genes. Involvement of some transcription factors in protein secretion was examined more in detail by means of gene disruption or gene up-regulation. In order to have a more clear understanding of the effect of these TFs in enhancement of protein secretion, a comparison with data previously obtained (see below) is suggested.

MAJOR COMMENTS

Lines 124-127. Analysis performed is unclear, please add more details here or at least in the Methods section. How did you calculate the scores for the transcription factors? How did you select the more relevant ones for further investigation?

Line 176-178. In some mutants the chr III is duplicated and this affected a large number of genes (e.g. TUP1). Did you take into account this problem in the evaluation of the genes identified? How did you compensate for the overexpression of genes localized on chr III?

Line 203-206. From the mutations identified in previous studies (Huang et al., PNAS 2015) is it possible to identify mutations involved in determining altered expression level of the genes identified in the present study? For examples, is there a possible explanation for the altered expression level of the hexose transporter genes? Is it possible to identify the putative mutation involved in determining the increased protein secretion? For example, in the previous paper it was reported a detailed description of the genes involved in UPR and in HSR; in the present paper again is reported a description of the genes involved in the UPR (line 271 and following) but it is unclear why authors did not commented results in light of the previous findings. For this reason, despite results reported are very interesting, it is very difficult to identify (or at least to suggest) the specific genetics determinant(s) of the observed phenotypes. One possible suggestion would be to perform a detailed investigation of the more interesting mutant(s) and to map the phenotypic trait by means of next generation assisted bulk segregant analysis. This suggestion is also particularly true if you consider what you reported at lines 352-355. The identification of the mutation determining the increased protein expression will help in the interpretation of the gene expression results and will improve the strategy used to select the more interesting mutations.

Another possible alternative approach could be to perform an in-silico evaluation of the mutations involved in the previous study by means of genome-scale metabolic models. This will allow a more accurate ranking of the mutations and the identification of those specifically involved in determining the increased protein secretion.

Line 274. Involvement of ERO1 gene in enhanced protein secretion was also found in a previous paper (Alane E. Wentz and Eric V. Shusta A Novel High-Throughput Screen Reveals Yeast Genes That Increase Secretion of Heterologous Proteins). In this paper a high-throughput screening strategy was used to identify yeast genes that increase secretion of heterologous proteins. The strategy is based on rapid flow cytometric screening of engineered yeast for gene products that improve the display of

heterologous proteins, but results are not compared to those reported in the present (nor in previous) paper(s).

Line 332. Here it is hard to understand if there is a significant correlation between the genes found differentially expressed and those regulated by the TF. How many genes regulated by a specific TF are differentially expressed in comparison to the total number of genes regulated by the TF itself? Is it possible to calculate a significance level?

MINOR COMMENTS

Line 53. Please modify the expression "Several mutant strains" with a more specific expression.

Line 93. Please add the names of the groups in supplementary figure 2 in order to simplify the reading.

Line 106. Please add the names of the groups in figure 2 in order to simplify the reading.

Figure 4. Please add the names of the strains to the "color schemes" reporting the up/down regulation of the genes. This will make the figure more clear.

Lines 165-166. Please reformulate sentence to avoid repetition.

Line 200. Please reformulate sentence to avoid repetition.

Line 372. Please check the sentence and correct.

Line 415, 438, 441. Please check for typos and correct.

Reviewer #2 (Remarks to the Author):

This paper covers a systems biology evaluation of protein secretion in yeast. The authors use a previously developed panel of yeast strains that have a 5-fold range of α -amylase expression. The paper is mostly a systems biology analysis with hypotheses driven from RNA-seq experiments. Overall, this is a very superficial paper that hypothesizes, but does not test, important factors for protein secretion.

The choice of exponential phase profiling is curious and not described. Exponential growth ends before the 40-hour timepoint (as shown in Supplementary Information Figure 1a), however the vast majority of α -amylase production (shown in Figure 1a) happens after the 40 hour timepoint. Is exponential phase profiling appropriate here to see what is influencing protein production? Does the profile change prior to and after exponential phase? None of this was tested or discussed.

The choice of timepoint notwithstanding, the study is a relatively standard approach of transcriptomics analysis with important transcription factors extracted and a GO ontology analysis conducted (a PCA analysis is peripherally described in the supporting information)

This analysis leads to a hypothesis about amino acid balancing and thiamin biosynthesis. Thus, the paper is rather speculative and does not describe any follow-up confirmation experiments. Very little followup (other than a ROS test) was conducted. There is no forward engineering effort or knockout or overexpression of target study conducted.

Finally, it seems the paper and conclusions may be very limited in scope. Is the particular subset of yeasts tested in this paper only specific for α -amylase production? Are these strains good for other proteins? The cysteine requirement and finding suggests that the results here may be very narrowly focused on only one protein of overexpression. How will this experiment and results relate to the protein therapeutics that the authors are claiming in the abstract and intro?

Reviewer #3 (Remarks to the Author):

Efficient protein production by yeast requires global tuning of metabolism

Obviously high-level protein secretion in yeast is a process influenced by multiple genes, some of them quite unexpected. The transcriptional analysis of the paper shows surprising increase in genes involved in anaerobic metabolism that is long known for higher supply rates of energy (ATP) but at lower efficiency with respect to glucose consumption. This is in clear contrast to mammalian protein secretion where reduced glycolysis and associated lower production of lactic acid promote protein production. The downregulation of amino acid metabolism is unexpected in a first view but considering the amino acid supply in the medium either free or as BSA together with an increased expression of AA transporters makes sense. Downregulation of glycolysis enzymes seems contradictory to increased anaerobic metabolism but makes sense knowing the overcapacities and the experimentally confirmed metabolic control of glucose uptake (Stuttgart and Delft groups). Also the observed downregulation of thiamine TFs makes sense since this is likely reducing the degradation of some crucial AAs as Phe, Ile etc.

Overall I think this is a thorough analysis of processes related to protein secretion and the created knowledge seems generally applicable in various fields where protein secretion is of importance.

I have, however, some comments and critiques that should be considered before publication.

1. The correlation of rates could be shown much better in a plot, e.g. all specific rates versus growth rate. This would also show some deviations from the general tendency in a clearer way (l.67 ff. and Table 1).
2. The comparison of intracellular fraction of amylase with protein yield and/or production rate would be interesting, also best shown by a plot (e.g. intracellular % versus yield and/or rate).
3. Why should downregulation of nucleotide synthesis and phosphate metabolism be beneficial for protein overproduction? How is the high demand for RNA synthesis accomplished? Are salvage pathways more efficient? Was there any indication of changes in this kind of metabolism?
4. L. 184 ff. Was there any indication of changes in lipid metabolism? It was observed that lipid content is increasing with increasing secretions (e.g. Klein et. al, 2014, *Metabol Eng*, 21:34 and 2015, *J Ind Microbiol Biotechnol*, 42:453).
5. The authors suggest an increase in the fraction of AAs derived from the medium instead from de novo synthesis. Is the content of all AAs (free + BSA) clearly in excess compared to the required amounts, particularly concerning also Cys? All data necessary to make such a calculation are available in the paper.
6. L. 321 ff: I cannot follow completely the argument concerning the increased expression of glutamine/glutamate related genes. If the biosynthesis of AAs is reduced and the uptake increased, why is the uptake not following the actual demand? Then amino donors would not be required at a higher level. Gln/Glu are, however, also involved in nucleotide biosynthesis. Could this potentially explain the experimental findings?
7. The reported 10 fold higher Cys requirement for amylase compared to the cellular protein looks strange to me. Experimental determination of Cys content in proteins, particularly in whole biomass, is quite delicate. I am skeptical about the content of Cys in the biomass because it is not really clear to me who actually measured the amino acid content in the biomass. I assume it is based on the data of Oura (Oura, E. 1983. Biomass from carbohydrates. In H.-J. Rehm and G. Reed (ed.), *Biotechnology*. Verlag Chemie, Weinheim, Germany) that were also used by Gombert et al. (2001, *J. Bacteriol*, 183, 1441) and many other groups practicing metabolic network studies in *S. cerevisiae*. Contrary to the chemically stable AAs, Cys is difficult to determine quantitatively. A clear description and discussion is required. Otherwise the Cys discussion has to be changed.
8. The reduced oxidative stress could also be related to an improved supply of NADPH, e.g. by the pentose phosphate pathway, however, no expression changes are seen in the data. Since high activities of the PPP were reported for the major mammalian cell producer CHO, it might be worth to

check this, by a simplified labelling method (e.g. Velagapudi et al., 2007, J Biotechnol 132:395)

9. Typos:

l.172: identified in this

l.207: this is was ??

l. 372: are listed in Suppl....

caption figure 1, l.5: remove "by"

caption figure 3: I see 6 scored reporter TFs in parts a and b.

Point by point response to Reviewers' comments:

We would like to thank the reviewers for their critical and positive evaluation and valuable comments to our manuscript. Their valuable comments helped us to improve the quality and the significance of our work, and we have revised the manuscript accordingly.

Please find our point-by-point responses to the reviewers' comments right after their original questions below:

Reviewer #1 (Remarks to the Author):

Huang and colleagues used genome-wide transcriptional analysis to investigate *Saccharomyces cerevisiae* mutants derived from the “strain AAC” and having increased efficiency in secretion of recombinant proteins. Results were validated using inverse engineering and physiological characterization. The paper is clear and well-written, some typos are present in the Methods section and a careful revision is suggested. The mutant strains investigated here are described in a paper previously published by the same authors. In this previous paper yeast mutants were analyzed using high throughput sequencing and a number of mutations on each strain were identified. However, there is very minor connection between the results reported in these two papers, while a more thorough analysis and a more clear comparison could provide a better comprehension of the results obtained. Additionally, in the present paper authors investigated the transcriptional regulatory responses in the mutant strains and integrated the transcriptome data with a network of transcription factors and associated genes. Involvement of some transcription factors in protein secretion was examined more in detail by means of gene disruption or gene up-regulation. In order to have a more clear understanding of the effect of these TFs in enhancement of protein secretion, a comparison with data previously obtained (see below) is suggested.

MAJOR COMMENTS

Lines 124-127. Analysis performed is unclear, please add more details here or at least in the Methods section. How did you calculate the scores for the transcription factors? How did you select the more relevant ones for further investigation?

Thank you for the good suggestion. More details about how to perform reporter analysis and how to calculate the scores for TFs and GO terms have been added to the method section. Transcription factors selected for further investigation were based on: 1) the TFs showing significant influence on its target genes in most of the strains (Figure 3a); 2) the TFs having significant expression changes themselves in most of the strains (Figure 3b); 3). the TFs that were not yet tested for their impact on protein secretion by deletion or overexpression previously.

Line 176-178. In some mutants the chr III is duplicated and this affected a large number of genes (e.g. TUP1). Did you take into account this problem in the evaluation of the genes identified? How did you compensate for the overexpression of genes localized on chr III?

The chr III was found to be duplicated in mutant strains MH34, D5, B130 and B184. The performance (higher protein secretion) of these four mutant strains was shaped not only by mutant genes but also by the duplicated chr III. The impact of a mutant gene on strain phenotype was mainly through alternation of its coding protein, and the expression level of a mutant gene itself may not change. In contrast, the impact of chr III duplication on strains was through directly elevated expression level of genes localized on the chr III. Both impact of mutant genes and impact of duplicated chr III on mutant strains were reflected by global transcriptional regulatory responses, which was unraveled by our reporter TF analysis. Evaluation of identified TFs aimed to stimulate transcriptional regulatory responses of mutant strains; regardless whether the transcriptional regulatory response was caused by mutant genes or by duplication of chr III. Our analysis therefore capture both the effect of chr III duplication and mutation in genes on transcriptional reprogramming. Our analysis can, however, not capture if there is altered activity in any of the proteins due to a mutation, but this would require a different type of analysis which is outside the scope of this study.

The differential expression level of genes was one of the inputs for the reporter analysis. We wondered if elevated expression level of genes on the duplicated chromosome may interfere analysis of global transcriptional regulatory responses. In order to study whether expression level of genes on the duplicated chr III influenced the reporter analysis, we did a reporter TFs analysis where we removed all genes on the chr III from the analysis, and then redid the reporter TF analysis for the strains MH34, D5, B130 and B184. As shown in the heat map below, identified TFs by the reporter analysis without including genes on chr III were almost the same as the identified TFs by the reporter analysis using data for all genes (Figure 3a), although the significances were different in these two reporter analysis. Only *SNF2* is not identified in the reporter analysis not including genes on chr III. This suggests that key information in global transcriptional regulatory responses, caused by mutations and chr III duplication, was not exclusively determined

by expression level of genes on chr III. In fact, not all genes on chr III have 2-fold expression level ($\text{Log}_2\text{Foldchange} = 1$), reflecting that their expression levels were not just simply determined by chr III duplication but had controlled coordinated expression, e.g. through feedback regulation. Hence, compensation for the overexpression of genes localized on chr III may be less important in evaluation of identified TFs. We have added a comment about this in the manuscript and we have added the figure below as Supplementary Figure 5.

Line 203-206. From the mutations identified in previous studies (Huang et al., PNAS 2015) is it possible to identify mutations involved in determining altered expression level of the genes identified in the present study? For examples, is there a possible explanation for the altered expression level of the hexose transporter genes? Is it possible to identify the putative mutation involved in determining the increased protein secretion? For example, in the previous paper it was reported a detailed description of the genes involved in UPR and in HSR; in the present paper again is reported a description of the genes involved in the UPR (line 271 and following) but it is unclear why authors did not commented results in light of the previous findings. For this reason, despite results reported are very interesting, it is very difficult to identify (or at least to suggest) the specific genetics determinant(s) of the observed phenotypes. One possible suggestion would be to perform a detailed investigation of the more interesting mutant(s) and to map the phenotypic trait by means of next gen-assisted bulk segregant analysis. This suggestion is also particularly true if you consider what you reported at lines 352-355. The identification of the mutation determining the increased protein expression will help

in the interpretation of the gene expression results and will improve the strategy used to select the more interesting mutations. Another possible alternative approach could be to perform an in-silico evaluation of the mutations involved in the previous study by means of genome-scale metabolic models. This will allow a more accurate ranking of the mutations and the identification of those specifically involved in determining the increased protein secretion.

Bulk segregant analysis is used to identify genetic markers (mutants) associated with a trait. In our previous study (PNAS 2015, 112:E4689–E4696), UV mutagenesis strains with higher protein secretion trait were segregated by microfluidic screening. We attempted to figure out if there was any consistent mutant gene within these mutant strains by whole genome sequencing. However, except inherited mutations from the parental strain, each mutant strain in the same generation has its own mutations, few common mutations were presented across the mutant strains. As protein secretion involves numerous processes, higher protein secretion may be affected not only by the direct process that a mutant gene is involved in, but also secondary cellular responses to the appearing mutation. Hence, the rationale for this study, that even though the mutant strains had many different mutations, mutant strains with higher protein secretion may have similar transcriptional regulatory responses caused by these different mutations. This hypothesis was confirmed by the present work, which mainly focused on transcriptional responses to the mutations and therefore has less emphasis on actual mutations. We have now clarified this in the introduction of the paper.

We agree with the reviewer that closer links of results (transcriptome changes) in the present study with results (mutations) in the previous study could provide more useful information. To establish links between mutations and differential expressed genes, we performed a GO slim mapper analysis (SGD, Yeast GO-Slim: Process) of differential expressed genes and mutant genes. Detailed mutant gene lists, percentage of mutant genes and percentage of differential expressed genes in each GO term are summarized in the new Table S2. This clearly shows how many mutant genes and what mutant genes there are in each GO term, and at the same time how many genes being differentially expressed are in each GO term. It was possible to point to mutations involved in determining altered expression level of the genes and increased protein secretion. For example, altered expression level of the hexose transporter genes, which are associated with the GO term carbohydrate transport, was likely affected by the mutant gene *SNF3*, which also belonged to the GO term carbohydrate transport, and that was involved in regulation of expressing *HXT* genes. In our previous study, we discussed the importance of UPR and HSR in protein secretion. Here, we found more evidences for altered activity of the UPR and HSR from transcriptional changes and by experimental validation (ROS level) to show reduced oxidative stress in the mutant strains. Hence, mutant genes and duplicated genes related to ER or organelle transport (such as *EMC1*, *ERV29*, *GOS1*,

USO1, *PDII*, *TDA3*, *VPS5*, *SNC2*) may contribute to reduced oxidative stress and facilitate improved protein secretion. We added comments about this in the text based on the new Table S2.

Line 274. Involvement of *ERO1* gene in enhanced protein secretion was also found in a previous paper (Alane E. Wentz and Eric V. Shusta A Novel High-Throughput Screen Reveals Yeast Genes That Increase Secretion of Heterologous Proteins). In this paper a high-throughput screening strategy was used to identify yeast genes that increase secretion of heterologous proteins. The strategy is based on rapid flow cytometric screening of engineered yeast for gene products that improve the display of heterologous proteins, but results are not compared to those reported in the present (nor in previous) paper(s).

Wentz et al. reported a useful method, which combined yeast surface display with rapid flow cytometric sorting, to screen yeast cDNA overexpression libraries for increasing protein secretion. In their study, overexpression of *ERO1* was found to improve antibody display/secretion. Their findings revealed the importance of protein folding in protein production, which was also shown in our study. We added a comment about the findings reported by Wentz et al. and discussed it in context of our data.

Line 332. Here it is hard to understand if there is a significant correlation between the genes found differentially expressed and those regulated by the TF. How many genes regulated by a specific TF are differentially expressed in comparison to the total number of genes regulated by the TF itself? Is it possible to calculate a significance level?

As suggested, we calculated the proportion of differentially expressed genes ($P\text{-adj} < 0.05$) out of the total number of genes regulated by *ROX1* in the mutant strains. The significance level of differential expression on the gene set level (genes regulated by *ROX1*) was calculated by using the Platform for Integrative Analysis of Omics (PIANO) R package. As shown in the left panel of the figure below, differential expression of the *ROX1* gene set (genes regulated by *ROX1*) was statistically significant ($P\text{-adj} < 0.01$) in the mutant strains. Similarly, we analyzed genes regulated by *HAPI* and the differential expression of *HAPI* gene set (genes regulated by *HAPI*) was statistically significant ($P\text{-adj} < 0.01$) in the mutant strains as well. So we are quite confident that the PIANO method does provide biological insight as also shown in many other studies.

MINOR COMMENTS

Line 53. Please modify the expression “Several mutant strains” with a more specific expression.

“several mutant strains” has been corrected as “seven mutant strains”

Line 93. Please add the names of the groups in supplementary figure 2 in order to simplify the reading.

The names of the groups have been added to the supplementary figure 2 (now the new supplementary figure 3). More details have been added to the figure legend as well.

Line 106. Please add the names of the groups in figure 2 in order to simplify the reading.

The names of the groups have been added to figure 2.

Figure 4. Please add the names of the strains to the “color schemes” reporting the up/down regulation of the genes. This will made the figure more clear.

The names of the strains have been provided in the lower-right corner of the figure.

Lines 165-166. Please reformulate sentence to avoid repetition.

The sentence has been revised as “To validate if the identified reporter TFs represent key points in regulation of protein secretion, some of the TFs were selected for evaluation.”

Line 200. Please reformulate sentence to avoid repetition.

The sentence has been revised as “The expression levels of genes were presented on a map of the central carbon metabolism.”

Line 372. Please check the sentence and correct.

“...listed Supplementary...” has been corrected as “...listed in Supplementary...”

Line 415, 438, 441. Please check for typos and correct.

Errors have been corrected.

Reviewer #2 (Remarks to the Author):

This paper covers a systems biology evaluation of protein secretion in yeast. The authors use a previously developed panel of yeast strains that have a 5-fold range of α -amylase expression. The paper is mostly a systems biology analysis with hypotheses driven from RNA-seq experiments. Overall, this is a very superficial paper that hypothesizes, but does not test, important factors for protein secretion.

The choice of exponential phase profiling is curious and not described. Exponential growth ends before the 40-hour timepoint (as shown in Supplementary Information Figure 1a), however the vast majority of α -amylase production (shown in Figure 1a) happens after the 40 hour timepoint. Is exponential phase profiling appropriate here to see what is influencing protein production? Does the profile change prior to and after exponential phase? None of this was tested or discussed.

The choice of timepoint notwithstanding, the study is a relatively standard approach of transcriptomics analysis with important transcription factors extracted and a GO ontology analysis conducted (a PCA analysis is peripherally described in the supporting information)

Thank you for your valuable comments. We agreed that we should describe more clearly why exponential phase profiling was chosen for our analysis. During the exponential phase, both carbon and nitrogen sources are in excess to support yeast cells growth, and in this phase the cells are in homeostasis. This phase is therefore commonly chosen by researchers to investigate yeast metabolism. Furthermore, we found previously that yeast

cells having increased secretion in the exponential phase have higher protein production at the end. In these studies (FEMS Yeast Research 2015, fov070; Appl. Microbiol. Biot. 2013, 97:3559-3568; BMC Biology 2012, 10:16.), transcriptional analysis of yeast cells in the exponential phase revealed important mechanisms that could be used for improving protein production.

To demonstrate this we calculated the relative amylase yield for the different strains in the exponential phase and at the end of the cultivations. This analysis shows that the mutant strains shows higher amylase yield compared to the reference strain AAC not only at the end, but also in the exponential phase (new Supplementary Figure 2a and b). We are therefore confident that it is possible to reveal important factors influencing protein secretion by profiling the cells in the exponential phase. Furthermore, in our study we experimentally validated hypothesis generated from our reporter TF analysis (Figure 3c and 3d).

It should be mentioned that gene expression levels could be different in other periods of the fermentation than in the exponential growth phase. We therefore agree with the reviewer that gene expression levels of different periods may provide additional information about the protein secretion process, but this is beyond the scope of this study. But we have added a comment about this as a direction of future studies in the discussion section.

This analysis leads to a hypothesis about amino acid balancing and thiamin biosynthesis. Thus, the paper is rather speculative and does not describe any follow-up confirmation experiments. Very little followup (other than a ROS test) was conducted. There is no forward engineering effort or knockout or overexpression of target study conducted.

We believe we did quite extensive experimental follow up studies to confirm our findings. We performed knockout or overexpression of reporter TFs to confirm their influence on protein production (Figure 3c and 3d). Protein production affected by amino acid balancing and thiamin biosynthesis were also validated through dedicated experiments (Figure 5c, 5e and 5f). Findings from our systems biology analysis were generally in

good agreement with physiological characterization of the mutant strains (Table 1, Table S1).

Finally, it seems the paper and conclusions may be very limited in scope. Is the particular subset of yeasts tested in this paper only specific for α -amylase production? Are these strains good for other proteins? The cysteine requirement and finding suggests that the results here may be very narrowly focused on only one protein of overexpression. How will this experiment and results relate to the protein therapeutics that the authors are claiming in the abstract and intro?

In our previous study (PNAS 2015, 112:E4689–E4696), production of the *Rhizopus oryzae* glucan-1,4- α -glucosidase was tested with the mutant strains and increased secretion of the enzyme suggested that the findings on α -amylase may be generally applicable. Here, we tested other two proteins, human serum albumin (HSA) and *Trichoderma reesei* endo-1,4-beta-xylanase II (XYN), in the mutant strain B184. Compared with the reference strain CEN.PK 530.1C, both proteins showed higher protein yield in B184 (new Supplementary Figure 9). These results show that our findings are not limited to a specific protein and also validated our findings with a pharmaceutical protein (HSA). As *S. cerevisiae* is both an industrially important cell factory for recombinant protein production and a key eukaryal model organism, our findings can most likely be used as guidelines for design of other cell factories for efficient pharmaceutical protein secretion.

Reviewer #3 (Remarks to the Author):

Efficient protein production by yeast requires global tuning of metabolism. Obviously high-level protein secretion in yeast is a process influenced by multiple genes, some of them quite unexpected. The transcriptional analysis of the paper shows surprising increase in genes involved in anaerobic metabolism that is long known for higher supply rates of energy (ATP) but at lower efficiency with respect to glucose

consumption. This is in clear contrast to mammalian protein secretion where reduced glycolysis and associated lower production of lactic acid promote protein production. The downregulation of amino acid metabolism is unexpected in a first view but considering the amino acid supply in the medium either free or as BSA together with an increased expression of AA transporters makes sense. Downregulation of glycolysis enzymes seems contradictory to increased anaerobic metabolism but makes sense knowing the overcapacities and the experimentally confirmed metabolic control of glucose uptake (Stuttgart and Delft groups). Also the observed downregulation of thiamine TFs makes sense since this is likely reducing the degradation of some crucial AAs as Phe, Ile etc.

Overall I think this is a thorough analysis of processes related to protein secretion and the created knowledge seems generally applicable in various fields where protein secretion is of importance.

I have, however, some comments and critiques that should be considered before publication.

1. The correlation of rates could be shown much better in a plot, e.g. all specific rates versus growth rate. This would also show some deviations from the general tendency in a clearer way (1.67 ff. and Table 1).

Thank you for this excellent comment. As suggested, the correlation of rates has been shown in plots (new Supplementary Figure 1a-c).

2. The comparison of intracellular fraction of amylase with protein yield and/or production rate would be interesting, also best shown by a plot (e.g. intracellular % versus yield and/or rate).

As suggested, the amylase yield was plotted versus intracellular fraction of amylase (%). The amylase yield showed a negative correlation with the intracellular fraction of amylase (new Supplementary Figure 1d).

3. Why should downregulation of nucleotide synthesis and phosphate metabolism be beneficial for protein overproduction? How is the high demand for RNA synthesis accomplished? Are salvage pathways more efficient? Was there any indication of changes in this kind of metabolism?

Protein overproduction usually causes metabolic burden and stress to cells (J. Biotechnol. 2004, 113:121-135.). Consequently, cells have to coordinate cellular processes to meet the demands of protein overproduction. In many cases, resources for cellular metabolism of cells are redirected to protein production and reduced biomass is observed. *De novo* nucleotide synthesis requires amino acids as substrate and is energy consuming processes. Increased protein production may be obtained through saving resources by having a reduced *de novo* nucleotide synthesis. As phosphate participates in nucleotide synthesis, downregulation of phosphate metabolism may occur due to the downregulation of nucleotide synthesis. The first two steps of *de novo* synthesis of pyrimidine are catalyzed by Ura2p, which is feedback inhibited by the level of UTP. The *URA2* gene was found to be downregulation ($P\text{-adj} < 0.05$) in our mutant strains. *FURI*, which encodes for an enzyme that converts uracil to UMP in the salvage pathways, was found to be upregulated in the mutant strains as well. These findings suggested that increasing demand for RNA in the mutant strains may be supported by increased salvage pathways. We have added a comment about this in the paper.

4. L. 184 ff. Was there any indication of changes in lipid metabolism? It was observed that lipid content is increasing with increasing secretions (e.g. Klein et. al, 2014, *Metabol Eng*, 21:34 and 2015, *J Ind Microbiol Biotechnol*, 42:453).

Klein et al. showed that the increased lipid content was most likely linked to an increasing demand for membranes and transport vesicles in the secretory pathway (*Metab. Eng.* 2014, 21:34-45.). Furthermore, vesicular trafficking pathways and membrane-bound signaling are regulated by protein lipidation, which is affected by lipid metabolism (*Curr. Opin. Chem. Biol.* 2015, 24:48-57.). In our study, GO terms “protein lipidation” and “lipid metabolic process” were found to be up-regulated. These findings indicated that

lipid metabolism changed in the mutant strains, and changes in lipid metabolism may contribute to the increased protein secretion. We have added a comment about this in the paper.

5. The authors suggest an increase in the fraction of AAs derived from the medium instead from de novo synthesis. Is the content of all AAs (free + BSA) clearly in excess compared to the required amounts, particularly concerning also Cys? All data necessary to make such a calculation are available in the paper.

Total AAs (free+BSA) added to the medium are 5.25 g/L, and Cys is 63.7 mg/L. According to the table below (new Table S3), supplied AAs were enough for the required amounts. Decreased de novo synthesis of AAs can probably save resources (intermediates and energy) to meet increased demands for amylase secretion.

Strain	Cell protein and amylase (g/L)	Cysteine content in cell protein and amylase (mg/L)
AAC	2.49 ± 0.25	5.7 ± 0.5
M715	2.28 ± 0.06	5.5 ± 0.2
MH23	2.22 ± 0.02	5.9 ± 0.1
MH34	2.30 ± 0.06	6.3 ± 0.4
F83	1.90 ± 0.09	5.4 ± 0.3
D5	2.38 ± 0.03	7.0 ± 0.2
B130	2.48 ± 0.19	7.7 ± 0.6
B184	2.35 ± 0.14	8.7 ± 0.5

6. L. 321 ff: I cannot follow completely the argument concerning the increased expression of glutamine/glutamate related genes. If the biosynthesis of AAs is reduced and the uptake increased, why is the uptake not following the actual demand? Then amino donors would not be required at a higher level. Gln/Glu are, however, also involved in nucleotide biosynthesis. Could this potentially explain the experimental findings?

As AAs were supplied in the medium, yeast cells were able to assimilate AAs directly for biosynthesis. Cells uptake AAs via AA transporters, some of which (e.g. *GAPI*) show broad substrate range. More than one kind of AAs can enter cells through the broad substrate range transporter. Endocytosis is another way for cells to take up AAs. Consequently, it may be difficult for cells to take up AAs just in actual stoichiometric demands. We found increased expression of genes involved in glutamate/glutamine biosynthesis and genes related to amino group transfer (transaminases). This suggested that conversion of AAs by amino group transfer was a possible solution for cells to balance the intracellular amino acids pool to what is required for protein production. Furthermore, as you mentioned, Gln and Glu are not just used as building blocks for

protein synthesis, they are also involve in many other processes (e.g. oxidative stress responses, nucleotide metabolism). We have added more discussion on this in the text.

7. The reported 10 fold higher Cys requirement for amylase compared to the cellular protein looks strange to me. Experimental determination of Cys content in proteins, particularly in whole biomass, is quite delicate. I am skeptical about the content of Cys in the biomass because it is not really clear to me who actually measured the amino acid content in the biomass. I assume it is based on the data of Oura (Oura, E. 1983. Biomass from carbohydrates. In H.-J. Rehm and G. Reed (ed.), *Biotechnology*. Verlag Chemie, Weinheim, Germany) that were also used by Gombert et al. (2001, *J. Bacteriol*, 183, 1441) and many other groups practicing metabolic network studies in *S. cerevisiae*. Contrary to the chemically stable AAs, Cys is difficult to determine quantitatively. A clear description and discussion is required. Otherwise the Cys discussion has to be changed.

Yes, the data of amino acid content in the yeast biomass (table below) reported by Oura (Oura, E. 1972. The effect of aeration on the growth energetics and biochemical composition of baker's yeast. PhD thesis, University of Helsinki, Finland.) were used in this study.

AAs	mmol/gDCW	AAs	mmol/gDCW	AAs	mmol/gDCW	AAs	mmol/gDCW
Ala	0.459	Gln	0.105	Leu	0.296	Ser	0.185
Arg	0.161	Glu	0.302	Lys	0.286	Thr	0.191
Asn	0.102	Gly	0.29	Met	0.051	Trp	0.028
Asp	0.297	His	0.066	Phe	0.134	Tyr	0.102
Cys	0.007	Ile	0.193	Pro	0.165	Val	0.265

Amino acid content in amylase and yeast presented as a percentage (in last version Table S2) may confuse readers. In order to compare the amino acid content in amylase with that in yeast in a clearer way, we calculated the amino acid composition of yeast cell protein based on the data of Oura and the coefficient of 0.45g protein/gDCW from reference by Forster et al. (*Genome Res.* 2003, 13: 244-253). This data was presented in the new Table S3 of revised manuscript, which contained a detailed table legend for readers following the calculation. The amino acid composition of amylase in mmol/g_{amylase} was calculated as well. In the new Table S3, different requirements of amino acid between production of 1 g amylase and production of 1 g yeast cell protein are clearly shown, and from here it is seen that there is a 9.3-fold requirement of Cys for synthesis of amylase compared with for synthesis of yeast cell protein. We revised the manuscript with a clear description and discussion accordingly.

8. The reduced oxidative stress could also be related to an improved supply of NADPH,

e.g. by the pentose phosphate pathway, however, no expression changes are seen in the data. Since high activities of the PPP were reported for the major mammalian cell producer CHO, it might be worth to check this, by a simplified labelling method (e.g. Velagapudi et al., 2007, J Biotechnol 132:395)

Besides CHO cell, the yeast *Pichia pastoris* was reported to benefit from increasing PPP for protein production (Appl. Microbiol. Biotechnol. 2016, 100:5955–5963). Improved supply of NADPH may be a general strategy to increase protein production. In our study, although no expression changes were seen in PPP, we found reduced biomass yield for the mutant strains (new Supplementary Figure 1e). A reduced biomass yield suggested that resources used for cellular synthesis were redirected to protein overexpression in the mutant strains. As many biosynthesis pathways require NADPH as reducing equivalents during cell growth, decreased biomass production is able to save NADPH that can be used for maintaining redox balance (protection against oxidative stress). Thus, the mutant strains seemed to meet the demands for NADPH in association with increased amylase production by redistribution of resources. This is further supported if we look at protein content in the mutant strains. As shown in the figure below (new Supplementary Figure 12), while the amylase content increased, the total protein content (yeast cell protein and amylase) didn't increase in the mutant strains compared with the reference strain AAC. Hence, the cells may not need to have increased PPP activity. However, as improved supply of NADPH is beneficial for protein production, it would be interested to metabolic engineer PPP in the mutant strains for protein production in future studies. We added a discussion about this to the text.

9. Typos:

1.172: identified in this

The error has been corrected

1.207: this is was ??

The error has been corrected

l. 372: are listed in Suppl....

The error has been corrected

caption figure 1, l.5: remove “by”

The error has been corrected

caption figure 3: I see 6 scored reporter TFs in parts a and b.

The reporter TFs analysis was used to calculate the significance for expression change of gene sets controlled by TFs, which were scored by the modulation in expression level of genes that controlled by TFs. TFs that received a median rank ≤ 5 (top 5), in distinct-directional up class and distinct-directional down class, were chosen and presented in Figure 3. We have included more details in the figure legend and method section to help readers better understand the results.

REVIEWERS' COMMENTS:

Reviewer #1 (Remarks to the Author):

The point by point response letter written by Huang and colleagues was examined as well as the modifications included in the revised version of the manuscript. Authors have carefully revised the manuscript according to the suggestions of the reviewers and performed additional experiments to address most of the criticisms raised. Specific analyses were included and described both in the manuscript and in additional information files and are relevant for a better understanding of the results reported. Some references were added in order to have a more complete comparison with results reported in previous studies dealing with mutants yeast strains characterized by increased protein secretion. From my point of view the manuscript has been substantially improved and worth to be published in Nature Communications.

Reviewer #2 (Remarks to the Author):

The authors have addressed most comments in the last review and this manuscript should now be acceptable for publication.

Reviewer #3 (Remarks to the Author):

I think the authors have carefully addressed the questions raised by reviewers thus improving the manuscript.

I do not have any further comments.

REVIEWERS' COMMENTS:

Reviewer #1 (Remarks to the Author):

The point by point response letter written by Huang and colleagues was examined as well as the modifications included in the revised version of the manuscript. Authors have carefully revised the manuscript according to the suggestions of the reviewers and performed additional experiments to address most of the criticisms raised. Specific analyses were included and described both in the manuscript and in additional information files and are relevant for a better understanding of the results reported. Some references were added in order to have a more complete comparison with results reported in previous studies dealing with mutants yeast strains characterized by increased protein secretion. From my point of view the manuscript has been substantially improved and worth to be published in Nature Communications.

Reviewer #2 (Remarks to the Author):

The authors have addressed most comments in the last review and this manuscript should now be acceptable for publication.

Reviewer #3 (Remarks to the Author):

I think the authors have carefully addressed the questions raised by reviewers thus improving the manuscript.

I do not have any further comments.

Response to Reviewers' comments:

We would like to thank the reviewers for useful comments and suggestions to improve our manuscript.